# Applying Natural Language Processing to Single-Report Prediction of Metastatic Disease Response Using the OR-RADS Lexicon

**DOI:** 10.3390/cancers15204909

**Published:** 2023-10-10

**Authors:** Lydia Elbatarny, Richard K. G. Do, Natalie Gangai, Firas Ahmed, Shalini Chhabra, Amber L. Simpson

**Affiliations:** 1School of Computing, Queen’s University, Kingston, ON K7L 2N8, Canada; 18le7@queensu.ca; 2Department of Radiology, Memorial Sloan Kettering Cancer Center, New York, NY 10065, USA; gangain@mskcc.org (N.G.); ahmedf1@mskcc.org (F.A.); chhabras@mskcc.org (S.C.); 3Department of Biomedical and Molecular Sciences, Queen’s University, Kingston, ON K7L 2V7, Canada

**Keywords:** natural language processing, metastasis, radiology, computed tomography, disease progression

## Abstract

**Simple Summary:**

Lack of standardization among radiologists in writing radiological reports impacts the ability to interpret cancer response to treatment at a large-scale. This is an issue since large-scale data collection is necessary to generate Real World Evidence (RWE) towards understanding the effectiveness of cancer treatments and developing personalized patient treatment decisions. This study aims to examine the utility of applying natural language processing (NLP) for large-scale interpretation of disease response using the standardized oncologic response categories known as the OR-RADS to facilitate RWE collection. This study demonstrates the feasibility of applying NLP to predict disease response in cancer patients, exceeding human performance, thus encouraging use of the standardized OR-RADS categories among radiologists and researchers to improve large-scale response prediction accuracy.

**Abstract:**

Generating Real World Evidence (RWE) on disease responses from radiological reports is important for understanding cancer treatment effectiveness and developing personalized treatment. A lack of standardization in reporting among radiologists impacts the feasibility of large-scale interpretation of disease response. This study examines the utility of applying natural language processing (NLP) to the large-scale interpretation of disease responses using a standardized oncologic response lexicon (OR-RADS) to facilitate RWE collection. Radiologists annotated 3503 retrospectively collected clinical impressions from radiological reports across several cancer types with one of seven OR-RADS categories. A Bidirectional Encoder Representations from Transformers (BERT) model was trained on this dataset with an 80–20% train/test split to perform multiclass and single-class classification tasks using the OR-RADS. Radiologists also performed the classification to compare human and model performance. The model achieved accuracies from 95 to 99% across all classification tasks, performing better in single-class tasks compared to the multiclass task and producing minimal misclassifications, which pertained mostly to overpredicting the equivocal and mixed OR-RADS labels. Human accuracy ranged from 74 to 93% across all classification tasks, performing better on single-class tasks. This study demonstrates the feasibility of the BERT NLP model in predicting disease response in cancer patients, exceeding human performance, and encourages the use of the standardized OR-RADS lexicon to improve large-scale prediction accuracy.

## 1. Introduction

Medical imaging, predominately computed tomography (CT), is an indispensable tool used to identify sites of active primary and metastatic disease and is performed routinely on cancer patients to monitor disease progression. Radiological reports are generated from the interpretation of CT imaging by radiology domain experts, summarizing disease findings and providing an overall impression. From these reports, extensive information on both patients’ historical clinical course and current disease status over prolonged periods of time can be extracted; this information can be used to generate Real World Data (RWD). The collection of RWD contributes to the generation of Real World Evidence (RWE), which is crucial for understanding the effectiveness of different cancer treatments and their effect and suitability for different patient subgroups [1]. However, since radiological reports are generally produced in unstructured, or in best cases, semi-structured free-text format, they typically require a trained human reader to interpret their findings [2,3,4], limiting their capacity for re-use throughout research and data mining [4], which presents several challenges.

Manual review of radiological reports for evaluating disease progression, response to therapy, and informing therapeutic decision-making is incredibly time-consuming, costly, and not feasible at a large scale [5,6]. The absence of a standardized lexicon or documentation practices throughout radiological reporting is an added challenge; this leaves radiologists to describe disease burden subjectively and qualitatively, increasing ambiguity and the potential for miscommunication [7,8,9]. Natural Language Processing (NLP) provides a solution to these challenges by enabling computers to derive meaning from natural language input through the conversion of unstructured text into a structured representation. Hence, it is increasingly being used to facilitate automatic identification and extraction of data from radiology reports [10]; however, its application in the classification of disease response from radiology reports has been limited [11].

Previously, the feasibility of applying NLP to individual radiology reports to assess spatial and temporal patterns of metastatic spread at a large scale was assessed [12]. This was achieved by comparing three approaches to creating NLP models and identifying that a term frequency–inverse document frequency (TF-IDF) model performed best. Extending on this, consecutive multi-report prediction using convolutional and recurrent neural network approaches demonstrated that multi-report NLP models predict the presence of metastatic disease with higher accuracy, precision, and recall compared to single-report prediction [13]. The feasibility of applying NLP to extract outcomes from oncologist notes was achieved using an NLP model to identify the presence of an assessment/plan section within the notes and then convolutional neural networks to assess the section for whether cancer was present/absent, progressing/worsening, and responding/improving [11]. These studies support NLP as an effective tool for the prediction of disease response; however, the challenge of lack of standardization within radiologist description and interpretation of disease response from clinical reports remains.

Response Evaluation Criteria in Solid Tumors (RECIST) provides guidelines for assessing tumor response by evaluating target and nontarget lesions [14] but is not routinely used outside of clinical trials [15] and does not effectively eliminate subjectivity regarding the target lesion due to its reliance on human measurement, which can translate into discrepancies in interpreting disease response [16,17]. Hence, a group at the Memorial Sloan Kettering Cancer Center recently piloted a novel method, the Oncologic Response Lexicon (OR-RADS), which comprises the following seven distinct response categories: Or01 = decreased, Or02 = slightly decreased, Or03 = unchanged, Or04 = slightly increased, Or05 = increased, OrXE = equivocal, and OrXM = mixed [18]. This lexicon aims to reduce communication ambiguities and addresses deficiencies in RECIST, supporting its greater practicality for use in routine clinical practice.

A previous study demonstrated that Bidirectional Encoder Representations from Transformers (BERT) trained on structured reports mined using RECIST can reach human performance in predicting disease response from free-text reports [19], although they found that lexical complexity and semantic ambiguities within radiology reporting lowered both human and model performance. This study supports BERT as an effective NLP model for disease response prediction, and furthermore, pairing BERT with a reporting lexicon that addresses the challenges with RECIST, such as OR-RADS, may potentially further improve BERT’s performance. BERT has also been demonstrated to outperform the traditional NLP approach, TF-IDF, across several text-classification tasks and provide the noteworthy benefits of transfer learning, further supporting its suitability [20].

This research aims to examine the utility of NLP for large-scale interpretation of disease response using the OR-RADS response categories to facilitate the generation of RWD. The use of the OR-RADS lexicon is proposed to promote the standardization of reporting and achieve accurate automatic assessment of disease response. Improving the accuracy of assessing a patient’s response to treatment at a large scale will be crucial for generating RWE, which ultimately guides therapeutic decision making; informing the suitability of patients’ current regimens; and, more broadly, assessing cancer treatment effectiveness outside of clinical trials to improve routine care practice.

## 2. Materials and Methods

This retrospective study was undertaken with a waiver of informed consent approved by the Institutional Review Board at Memorial Sloan Kettering Cancer Center (MSK). The data consist of 3503 retrospectively collected single radiology reports from CT imaging studies across various cancer types performed at MSK within an 11-month time period. Of these reports, free-text clinical impressions summarizing important findings were extracted and manually labeled with one of the seven OR-RADS response categories (Figure 1) by board-certified attending radiologists. The OR-RADS category assigned by the radiologist for each CT at the time of clinical reporting is referred to as the “True Label”.

The first five response categories spanning from Or01 to Or05 describe the range of decreased (Or01) to increased disease (Or05) since the patient’s previous scan, with the Or03 category describing unchanged disease. The final two categories, OrXE and OrXM, describe equivocal progression and mixed response, respectively. An equivocal progression (OrXE) label is used when new imaging findings that may represent disease progression are present, yet some degree of uncertainty remains, and a mixed response (OrXM) label is used when a radiological report indicates both an unequivocal increase and decrease in disease extent either within the same and/or between different organs [18].

All free text data were originally collected in consistent uppercase, preserved in this format, and further normalized by removing numeric characters to ensure that term frequency was not impacted by lexical or stylistic differences. The impressions consistently included the date of the patient’s previous CT scan; thus, regular expressions were used to identify and remove these text patterns as they are not clinically relevant to the single-report findings. This also allows the vocabulary size to remain manageable.

The normalized free text data then underwent tokenization to encode them into the appropriate input format for the NLP model. Special tokens, [CLS] and [SEP], were added to make the encoding compatible with the BERT input format. The [CLS] token marks the start of a single text input, and the [SEP] token is added to mark the end of each sentence in the text input. These tokens are added automatically during the tokenization process. Additionally, the seven response category labels were transformed into one-hot encoded target values, wherein each label is represented by a vector of length seven. Each position in the vector signifies one of each of the seven response categories and is marked with a single 1 in that position, while the remaining vector positions are marked with 0s. Each text input encoding is then mapped to its target value.

This study employed a BERT model to achieve three classification objectives: 1. multiclass classification of the seven OR-RAD labels; 2. single-class classification of OR-RAD label 1 (Or01 = decreased); and 3. single-class classification of OR-RAD label 5 (Or05 = increased). Since disease response runs along a continuum, with the OR-RAD labels as interval checkpoints in this case, the multiclass classification task was performed to generate data capturing the full range of disease response possibilities across each radiological report. The two single-class classification tasks for Or01 and Or05 were then performed to identify “decreased” and “increased” disease burden, respectively, which are the most clinically relevant categories for the generation of RWE.

The BERT model developed was pretrained on the English language using the BERT-base uncased tokenizer. This model’s standard architecture comprises 12 transformer block layers with a hidden size of 768 and 12 self-attention heads (Figure 2). For each of the three classification objectives, the following consistent hyperparameters were used: maximum sequence length of 128, 10 epochs, batch size of 16, learning rate of 1 × 10^−5^, and dropout regularization with a probability of 0.4. Adam optimizer and cross-entropy loss function were used during training. The data were randomly separated into 80–20% split train and test sets (i.e., 80% training, 10% validation, and 10% test). Performance of the NLP model was measured using evaluation metrics of accuracy, precision, recall, and F1 scores, alongside evaluating, for each OR-RAD category, the levels of overprediction (the number of false-positive predictions) and underprediction (the number of false-negative predictions). Additionally, three radiologists were recruited to retrospectively label the test set to compare human performance with the NLP model performance. The test set was divided with no overlap among the radiologists for manual annotation. The same evaluation metrics were used to measure human performance.

## 3. Results

### 3.1. NLP Model Performance

The performance of the NLP model in the training, validation, and test sets for each of the three classification objectives is recorded in Table 1. Model accuracies were above 99% in the training set, above 95% in the validation set, and above 96% in the test set.

#### 3.1.1. Model Performance in Multiclass Classification of OR-RADS

The seven-class OR-RADS classification model achieved an accuracy of 95.2% in the validation set and 96.3% in the test set. The detailed metrics for the validation and test set are shown in Table 2 and Table 3 and Table 4 and Table 5, respectively. In the validation set (*n* = 351), the model predicted the Or05 and Or03 class reports most correctly and had relatively more difficulty predicting the OrXE class. OrXE and OrXM were the most overpredicted classes, and OrXE and Or02 were the most underpredicted classes. In the test set (*n* = 350), apart from classes Or01 and Or05, all five other classes were underpredicted only once. Or01 and Or05 were both underpredicted two times; however, the model produced no overpredictions for these two classes. OrXE and Or04 were the most overpredicted classes, and class Or01 was the most underpredicted class.

An example report that was misclassified by the model during the multiclass classification task is shown as follows in Figure 3:

#### 3.1.2. Model Performance in Single-Class Classification of OR-RAD Or01 = Decreased

The Or01 single-class classification model achieved an accuracy of 98.6% in the validation set and 98.0% in the test set. The detailed metrics for the validation and test set are shown in Table 6 and Table 7 and Table 8 and Table 9, respectively. In the validation set (*n* = 351), Or01 class reports were underpredicted three times and overpredicted two times, and in the test set (*n* = 350), they were underpredicted six times and overpredicted once.

An example report that was misclassified by the model during the Or01 single-class classification task is shown as follows in Figure 4. This report was also misclassified by the model during the multiclass classification task.

#### 3.1.3. Model Performance in Single-Class Classification of OR-RAD Or05 = Increased

The Or05 single-class classification model achieved an accuracy of 97.7% in the validation set and 98.9% in the test set. The detailed metrics for the validation and test set are shown in Table 10 and Table 11 and Table 12 and Table 13, respectively. In the validation set (*n* = 351), Or05 class reports were underpredicted five times and overpredicted three times, and in the test set (*n* = 350), they were underpredicted and overpredicted two times each.

An example report that was misclassified by the model during the Or05-single class classification task is shown as follows in Figure 5. This report was also misclassified by the model during the multiclass classification task.

### 3.2. Human Performance

#### 3.2.1. Human Performance in Multiclass Classification of OR-RADS

The radiologists conducting the seven-class OR-RADS classification task on the test set achieved an accuracy of 74.9.3%, substantially lower than the 96.3% accuracy achieved by the BERT model. The detailed metrics for the test set are shown in Table 14 and Table 15. Like the BERT model, the radiologists were able to predict Or03 and Or05 the most correctly. The radiologists had more difficulty predicting classes Or02, OrXM, and OrXE. Generally, Or02 class reports tended to be underpredicted as being either Or01 or Or03, and OrXE as either Or05 or Or03. Despite Or05 being one of the most correctly predicted classes, it was conversely the most overpredicted; generally, the misclassified Or04, OrXE, or OrXM class reports were most prone to being incorrectly classified as Or05.

An example report shown in Figure 3, which was misclassified by the NLP model during the multiclass classification task, was correctly classified by the radiologists. An example report that was misclassified by both the model and radiologists during this task is shown as follows in Figure 6.

#### 3.2.2. Human Performance on Single-Class Classification of OR-RAD Or01 = Decreased

The radiologists conducting the Or01 single-class classification task on the test set achieved an accuracy of 92.6% compared to the 98.0% accuracy of the BERT model. The detailed metrics for the test set are shown in Table 16 and Table 17. Or01 class reports were underpredicted 12 times and overpredicted 14 times.

The example report shown in Figure 4, which was misclassified by the NLP model during the Or01 single-class classification task, was correctly classified by the radiologists. An example report that was misclassified by both the model and radiologists during this task is shown as follows in Figure 7.

#### 3.2.3. Human Performance on Single-Class Classification of OR-RAD Or05 = Increased

The radiologists conducting the Or05 single-class classification task on the test set achieved an accuracy of 90.9% compared to the 98.9% accuracy of the BERT model. The detailed metrics for the test set are shown in Table 18 and Table 19. Or05 class reports were underpredicted 9 times and overpredicted 23 times.

The example report shown in Figure 5, which was misclassified by the NLP model during the Or05 single-class classification task, was correctly classified by the radiologists. An example report that was misclassified by both the model and radiologists during this task is shown as follows in Figure 8.

## 4. Discussion

In this study, a BERT model was developed for single-report prediction of metastatic disease response as categorized by the seven response labels defined in the OR-RADS lexicon. The key findings of the study were: (A) All three classification objectives that the model was applied in (i.e., seven-class classification and single-class classification of Or01 and of Or05) demonstrated high performance in accomplishing their classification tasks. (B) The model achieved higher performance in both the single-class objectives compared to the multiclass objective for disease response classification. (C) During testing, the model produced relatively few misclassifications overall. (D) The NLP model’s prediction capability outperformed that of the radiologists in all three classification tasks.

In the current study, BERT was applied to the encoded free text data, as Fink et al. had previously demonstrated BERT’s success in a similar response prediction application using RECIST. However, they also identified how reporting ambiguity using RECIST interfered with performance. Hence, we incorporated the OR-RADS lexicon, which was developed to address ambiguities with RECIST, to train the model in this study. Fink et al. reported performance metrics of accuracy, F1 score, precision, and recall ranging from 67 to 71.4% for their BERT model trained using RECIST data and metrics ranging from 73.5 to 74.2% for their human annotation by a radiologist. The current study reports metrics ranging from 94.8 to 96.3% for the BERT model’s performance in the multiclass classification task and 66.3 to 74.9% for the radiologist’s annotation in the same task. While the human performance reported by Fink et al. is similar to that of the current study, their BERT model’s performance is slightly lower than their human performance, whereas the model performance in the current study is notably higher than the human performance. In this retrospective study, the model’s resulting high prediction accuracy, which exceeded human performance, supports the use of BERT classification based on the OR-RADS response categories to help achieve better success, clarity, and standardization in response prediction.

The model performed best in the single-class classifications of Or-01 and Or-05 in comparison to the multiclass classification of the seven OR-RADS categories. This likely was due to the model being able to handle the decision boundaries of the single classes in isolation better than all seven classes simultaneously.

While the model produced relatively few misclassifications, there tended to be commonalities between the reported misclassifications. Throughout the multiclass classification task, the model had difficulty predicting the “equivocal” (OrXE) and “mixed” (OrXM) response reports, most frequently due to overpredicting them as other response labels. Clinically, this makes sense as these two response labels can contain elements of other labels that motivate their equivocal or mixed assignment. The radiologists also tended to have difficulty predicting those same two classes, as well as the “slightly decreased” (Or02) class reports. Next, throughout the Or01 single-class classification task, the model tended to underpredict the “decreased” (Or01) response reports, as it also did during the multiclass classification task.

In comparison to the model performance, human performance produced more frequent misclassifications. The radiologists achieved an almost equal number of over and underpredictions of class Or01 reports, but with double the amount of underpredictions and 13 more overpredictions compared to the model’s results. Finally, throughout the Or05 single-class classification task, the model underpredicted and overpredicted the “increased” (Or05) response reports each twice. The radiologists produced 7 more underpredictions and 21 more overpredictions compared to the model’s results. Ultimately, the NLP model provided better predictions in this dataset.

Despite the NLP model’s consistent outperformance, there were several reports that the model was unsuccessful in interpreting, but the radiologist’s annotation was correct. In fact, most reports that were misclassified by the model were correctly classified by the radiologist. This could be because those reports contained additional clinical terminology that is not relevant to the disease progression or response; for example, half of the report shown in Figure 5 discusses surgical removal procedures, such as: “interim cystoprostatectomy with ileal conduit”. These non-disease-related comments likely misled the model’s prediction, whilst the radiologists with domain expertise could interpret (and discard) them appropriately.

While, in these cases, radiology domain expertise was required to correctly interpret the clinical impression, there were several cases where human classification capabilities were insufficient. Commonly, incorrect radiologist annotations tended to be close to the correct label; for example, a report with the true label “decreased” (Or01) was predicted “slightly decreased” (Or02); alternatively, a report with a pure label (one of Or01 to Or05) was predicted “mixed” (OrXM). Additionally, OrXE, OrXM, and “slightly increased” (Or04) class reports were very commonly incorrectly classified as “increased” (Or05). There are several potential explanations for these incorrect human annotations. Human error caused by fatigue is potentially a factor during retrospective labeling. The imbalance of category labels in the dataset may have also introduced bias. It is also possible for there to have been a drift in the use of the OR-RADS categories by radiologists during the period in which these categories were first developed and used prospectively, compared to more recently at the time that the retrospective human annotations occurred.

Based on the currently generated data, there is evidence to support the value of using this model to conduct standardized, large-scale prediction of disease response by addressing the challenges of inconsistency and ambiguity in radiological reporting. It is recognized that there is still a time and cost investment in extracting clinical impression data from the radiological report for BERT classification. However, standardized determination of metastatic progression on a large scale is fundamental to assessing patients’ response to therapy at the population level. Furthermore, there is potential for this model’s use in large-scale epidemiological analysis of pathology and disease progression in various cancers across many patients. This would help generate RWE to support the evaluation of cancer treatment effectiveness and contribute towards improving routine care practice.

A limitation of the study is that the standard of reference for the raw labeled data used in model development was based only on the information provided in the free-text clinical impression. Due to limited access as well as the anticipated additional workload, the corresponding full-length radiological reports and CT images were not used to verify the model predictions, but these are potential areas of further investigation. Additionally, the data used in this study did not include further information on cancer type, therapies, or other patient demographics. Given the relatively small sample size in this study and the low failure rate compared to the total number of variables that would be introduced by including cancer subtypes and therapies, an investigation of the potential effects of these variables on misclassifications by the BERT model would be a future potential area of research.

In future work, increasing the dataset of radiological impression data would expand the scope of classification and improve the outcomes of the study. It may also be beneficial to explore a synthetically generated or augmented dataset of radiological impressions to address any imbalances in the number of cases for each category label. Additionally, exploring the use of ensemble models, as well as other BERT models pretrained specifically on clinical terminology, may improve overall performance accuracy and address the difficulty in predicting the equivocal (OrXE) and mixed (OrXM) classes. There may also be merit in training a BERT model on radiological terminology specifically, which may address the current model’s difficulty in differentiating non-cancer-related clinical terms. Finally, it may be best to train a BERT model on OR-RADS data limited to certain primary cancers of interest, such as lung and colorectal cancer, before applying the model predictions to Real World Evidence research.

## 5. Conclusions

In conclusion, this study demonstrates the feasibility of the BERT NLP model to predict disease response and specifically encourages the use of the standardized OR-RADS lexicon in routine clinical reports to improve prediction accuracy at a large scale. This standardized prediction model shows promise as an automatic disease response assessment tool to aid in generating RWE towards evaluating treatment effectiveness to guide therapeutic decision making, inform patient regimen suitability, and ultimately, improve the standard of care.

## Figures and Tables

**Figure 1 cancers-15-04909-f001:**
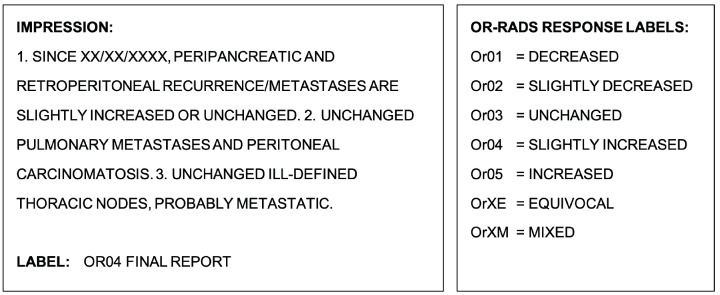
Example clinical impression extracted from a radiological report (**left**) and the reference OR-RADS disease response label definitions (**right**).

**Figure 2 cancers-15-04909-f002:**
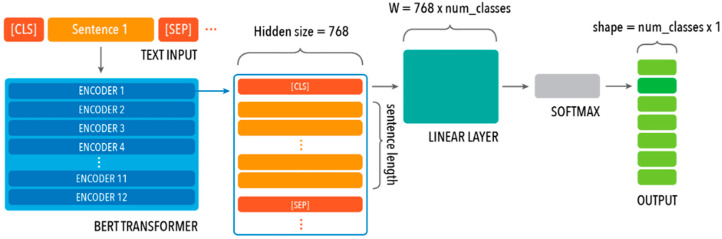
BERT base uncased model architecture, which comprises 12 transformer block layers, each with a hidden size of 768, and an added linear layer and softmax, which yields the output.

**Figure 3 cancers-15-04909-f003:**
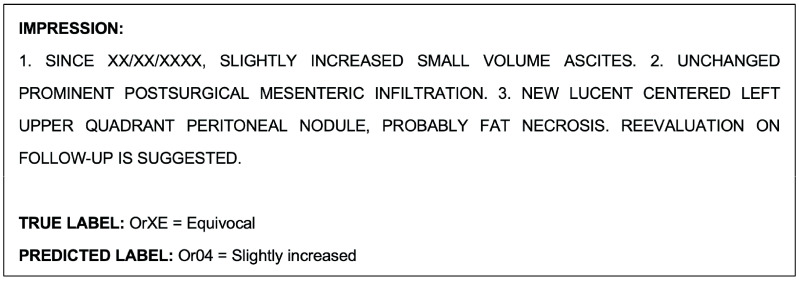
Example radiological report impression that was misclassified by the model during the multiclass classification task. The equivocal label was chosen because of the finding of a new peritoneal nodule, which was interpreted as probably fat necrosis, a benign process.

**Figure 4 cancers-15-04909-f004:**
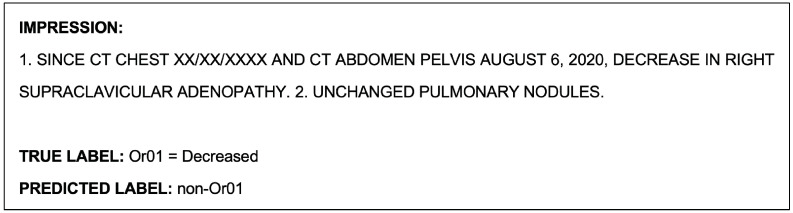
Example radiological report impression that was misclassified by the model during the Or01 single-class classification task.

**Figure 5 cancers-15-04909-f005:**
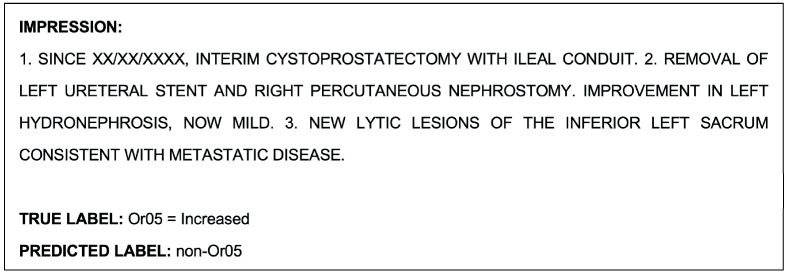
Example radiological report impression that was misclassified by the model during the Or05 single-class classification task.

**Figure 6 cancers-15-04909-f006:**
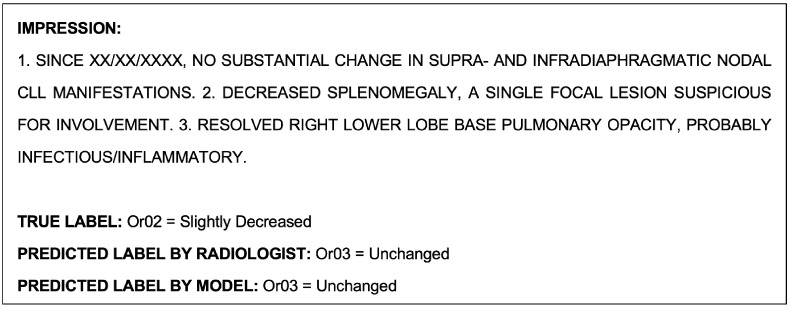
Example radiological report impression misclassified by both the model and radiologists during the multiclass classification task.

**Figure 7 cancers-15-04909-f007:**
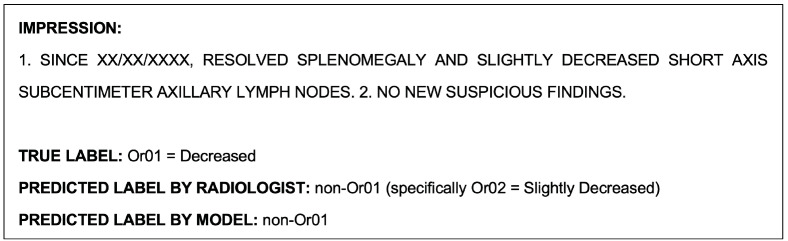
Example radiological report impression that was misclassified by both the model and radiologists during the Or01 single-class classification task.

**Figure 8 cancers-15-04909-f008:**
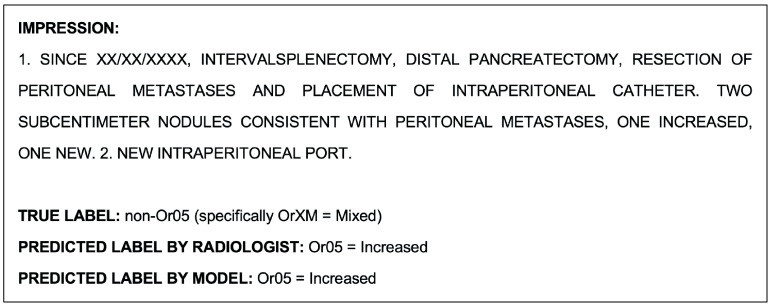
Example radiological report impression that was misclassified by both the model and radiologists during the Or05 single-class classification task.

**Table 1 cancers-15-04909-t001:** Overview of model performance on all three classification objectives. Performance is indicated by accuracy, F1 score, precision, and recall metrics for training (Train), validation (Val), and test sets.

		7-Class Classification of OR-RADS			Single-Class Classification of Or01			Single-Class Classification of Or05		
	Metrics	Train Set (2802)	Val Set (351)	Test Set (350)	Train Set (2802)	Val Set (351)	Test Set (350)	Train Set (2802)	Val Set (351)	Test Set (350)
	Accuracy	99.1%	95.2%	96.3%	99.7%	98.6%	98.0%	99.6%	97.7%	98.9%
Macro Average	F1 score	98.8%	92.6%	95.5%	99.4%	97.7%	96.6%	99.4%	97.0%	98.5%
	Precision	98.7%	92.2%	94.8%	99.3%	98.0%	98.1%	99.3%	97.3%	98.5%
	Recall	98.8%	93.3%	96.3%	99.5%	97.4%	95.2%	99.5%	96.6%	98.5%

Note: Data are percentages, with the numbers of reports in each set listed within parentheses. F1 Score, precision, and recall are macro averages. Val: Validation.

**Table 2 cancers-15-04909-t002:** Validation set model performance on the multiclass classification of OR-RADS.

Metrics	OR-RAD Labels						
	Or01 (68)	Or02 (28)	Or03 (92)	Or04 (29)	Or05 (90)	OrXE (27)	OrXM (17)
F1 score	97.0%	90.6%	97.8%	90.3%	98.3%	85.2%	88.9%
Precision	98.5%	96.0%	97.8%	84.8%	98.9%	85.2%	84.2%
Recall	95.6%	85.7%	97.8%	96.6%	97.8%	85.2%	94.1%
Accuracy	95.2%						

Note: Data are percentages, with the numbers of reports in each set listed within parentheses.

**Table 3 cancers-15-04909-t003:** Validation set model confusion matrix for the multiclass classification of OR-RADS.

		Predicted Classes						
		Or01	Or02	Or03	Or04	Or05	OrXE	OrXM
True Classes	**Or01**	65	1	0	0	0	2	0
	**Or02**	0	24	1	0	0	1	2
	**Or03**	0	0	90	1	0	1	0
	**Or04**	0	0	1	28	0	0	0
	**Or05**	0	0	0	1	88	0	1
	**OrXE**	1	0	0	2	1	23	0
	**OrXM**	0	0	0	1	0	0	16

Note: True predictions are highlighted in green.

**Table 4 cancers-15-04909-t004:** Test set model performance on the multiclass classification of OR-RADS.

Metrics	OR-RAD Labels						
	Or01 (65)	Or02 (31)	Or03 (75)	Or04 (31)	Or05 (87)	OrXE (36)	OrXM (25)
F1 score	95.2%	95.2%	98.0%	92.3%	98.8%	94.6%	94.1%
Precision	100.0%	93.8%	97.4%	88.2%	100.0%	92.1%	92.3%
Recall	90.8%	96.8%	98.7%	96.8%	97.7%	97.2%	96.0%
Accuracy	96.3%						

Note: Data are percentages, with the numbers of reports in each set listed within parentheses.

**Table 5 cancers-15-04909-t005:** Test set model confusion matrix for the multiclass classification of OR-RADS.

		Predicted Classes						
		Or01	Or02	Or03	Or04	Or05	OrXE	OrXM
True Classes	**Or01**	59	2	1	0	0	3	0
	**Or02**	0	30	1	0	0	0	0
	**Or03**	0	0	74	1	0	0	0
	**Or04**	0	0	0	30	0	0	1
	**Or05**	0	0	0	1	85	0	1
	**OrXE**	0	0	0	1	0	35	0
	**OrXM**	0	0	0	1	0	0	24

Note: True predictions are highlighted in green.

**Table 6 cancers-15-04909-t006:** Validation set model performance on the single-class classification of OR-RAD Or01. (Model performance metrics).

Metrics	OR-RAD Labels	
	Not Or01 (283)	Or01 (68)
F1 score	99.1%	96.3%
Precision	98.9%	97.0%
Recall	99.3%	95.6%
Accuracy	98.6%	

Note: Data are percentages, with the numbers of reports in each set listed within parentheses.

**Table 7 cancers-15-04909-t007:** Validation set model performance on the single-class classification of OR-RAD Or01. (Confusion matrix).

		Predicted Classes	
		Not Or01	Or01
True Classes	**Not Or01**	281	2
	**Or01**	3	65

Note: True predictions are highlighted in green.

**Table 8 cancers-15-04909-t008:** Test set model performance on the single-class classification of OR-RAD Or01 (Model performance metrics).

Metrics	OR-RAD Labels	
	Not Or01 (285)	Or01 (65)
F1 score	98.8%	94.4%
Precision	97.9%	98.3%
Recall	99.6%	90.8%
Accuracy	98.0%	

Note: Data are percentages, with the numbers of reports in each set listed within parentheses.

**Table 9 cancers-15-04909-t009:** Test set model performance on the single-class classification of OR-RAD Or01. (Confusion matrix).

		Predicted Classes	
		Not Or01	Or01
True Classes	**Not Or01**	284	1
	**Or01**	6	59

Note: True predictions are highlighted in green.

**Table 10 cancers-15-04909-t010:** Validation set model performance on the single-class classification of OR-RAD Or05 (Model performance metrics).

Metrics	OR-RAD Labels	
	Not Or05 (261)	Or05 (90)
F1 score	98.5%	95.5%
Precision	98.1%	96.6%
Recall	98.9%	94.4%
Accuracy	97.7%	

Note: Data are percentages, with the numbers of reports in each set listed within parentheses.

**Table 11 cancers-15-04909-t011:** Validation set model performance on the single-class classification of OR-RAD Or05 (Confusion matrix).

		Predicted Classes	
		Not Or05	Or05
True Classes	**Not Or05**	258	3
	**Or05**	5	85

Note: True predictions are highlighted in green.

**Table 12 cancers-15-04909-t012:** Test set model performance on the single-class classification of OR-RAD Or05 (Model performance metrics).

Metrics	OR-RAD Labels	
	Not Or05 (263)	Or05 (87)
F1 score	99.2%	97.7%
Precision	99.2%	97.7%
Recall	99.2%	97.7%
Accuracy	98.9%	

Note: Data are percentages, with the numbers of reports in each set listed within parentheses.

**Table 13 cancers-15-04909-t013:** Test set model performance on the single-class classification of OR-RAD Or05 (Confusion matrix).

		Predicted Classes	
		Not Or05	Or05
True Classes	**Not Or05**	261	2
	**Or05**	2	85

Note: True predictions are highlighted in green.

**Table 14 cancers-15-04909-t014:** Test set human performance on the human multiclass classification of OR-RADS.

Metrics	OR-RAD Labels						
	Or01 (65)	Or02 (31)	Or03 (75)	Or04 (31)	Or05 (87)	OrXE (36)	OrXM (25)
F1 score	80.3%	52.0%	86.1%	61.5%	83.0%	57.1%	52.6%
Precision	79.1%	68.4%	78.9%	76.2%	77.2%	80.0%	46.9%
Recall	81.5%	41.9%	94.7%	51.6%	89.7%	44.4%	60.0%
Accuracy	74.9%						

Note: Data are percentages, with the numbers of reports in each set listed within parentheses.

**Table 15 cancers-15-04909-t015:** Test set human confusion matrix regarding the multiclass classification of OR-RADS.

		Predicted Classes						
		Or01	Or02	Or03	Or04	Or05	OrXE	OrXM
True Classes	**Or01**	53	5	1	0	1	2	3
	**Or02**	8	13	6	2	0	0	2
	**Or03**	2	0	71	0	0	1	1
	**Or04**	0	0	4	16	7	0	4
	**Or05**	0	0	1	2	78	1	5
	**OrXE**	2	0	7	1	8	16	2
	**OrXM**	2	1	0	0	7	0	15

Note: True predictions are highlighted in green.

**Table 16 cancers-15-04909-t016:** Test set human performance on the single-class classification of OR-RAD Or01 (Human performance metrics).

Metrics	OR-RAD Labels	
	Not Or01 (285)	Or01 (65)
F1 score	95.4%	80.3%
Precision	95.8%	79.1%
Recall	95.1%	81.5%
Accuracy	92.6%	

Note: Data are percentages, with the numbers of reports in each set listed within parentheses.

**Table 17 cancers-15-04909-t017:** Test set human performance on the single-class classification of OR-RAD Or01 (confusion matrix).

		Predicted Classes	
		Not Or01	Or01
True Classes	**Not Or01**	271	14
	**Or01**	12	53

Note: True predictions are highlighted in green.

**Table 18 cancers-15-04909-t018:** Test set human performance on the single-class classification of OR-RAD Or05 (Human performance metrics).

Metrics	OR-RAD Labels	
	Not Or01 (285)	Or01 (65)
F1 score	95.4%	80.3%
Precision	95.8%	79.1%
Recall	95.1%	81.5%
Accuracy	92.6%	

Note: Data are percentages, with the numbers of reports in each set listed within parentheses.

**Table 19 cancers-15-04909-t019:** Test set human performance on the single-class classification of OR-RAD Or05. (confusion matrix).

		Predicted Classes	
		Not Or01	Or01
True Classes	**Not Or01**	271	14
	**Or01**	12	53

Note: True predictions are highlighted in green.

## Data Availability

Due to institutional regulation for the data transfer agreement, we are unable to make the dataset available online.

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
