# Peer review of "Applying Natural Language Processing to Single-Report Prediction of Metastatic Disease Response Using the OR-RADS Lexicon"

_cancers, 2023, doi:10.3390/cancers15204909_

Round 1

Reviewer 1 Report

(1)In Section 2, the author should use some figures (for architecture) and equations to specify the methodology.

(2)The figures for architecture should be clear. The details of the blocks should be clearly defined and explained directly in the figures even they have been explained in text. Please provide the information as much as possible, then the readers can get the details from each separate figure easily even without the text.

(3)The authors should provide all the parameters of the used models.

(4)In Table 1, the number of training sample is too large. Please show the results of the different ratios between the training to test sets.

(5)In my opinion, Table 5 contains Tables 6~9. Are Tables 6~9 necessary?

(6)In Section 4, it is better to specify the findings point-by-point.

(7)The authors just simply used the existing models, so the academic value is limited. It is better if the authors could improve the models.

None.

Author Response

Reviewer 1

INTRODUCTORY RESPONSE:

Thank you for your close evaluation of the manuscript and for highlighting the following areas of improvement which are addressed below.

(1)In Section 2, the author should use some figures (for architecture) and equations to specify the methodology.

RESPONSE:

A schematic diagram of the model architecture was developed and included in section 2.

(2) The figures for architecture should be clear. The details of the blocks should be clearly defined and explained directly in the figures even they have been explained in text. Please provide the information as much as possible, then the readers can get the details from each separate figure easily even without the text.

RESPONSE:

Specific labels were included in the architecture diagram (Figure 2) to aid in the comprehension of the BERT model. Additional detail is provided in the text (page 4).

(3)The authors should provide all the parameters of the used models.

RESPONSE:

The specific parameters of the model were added to section 2 (page 4).

(4)In Table 1, the number of training sample is too large. Please show the results of the different ratios between the training to test sets.

RESPONSE:

Thank you for this comment. Standard guidelines for reporting AI research, such as “Checklist for Artificial Intelligence in Medical Imaging (CLAIM)” and “Transparent Reporting of a multivariable prediction model for Individual Prognosis Or Diagnosis (TRIPOD)”, do not specify the need to study multiple split ratios. The train-validation-test split of 80%-10%-10% that was selected to be used in this study was informed by standard practice.  

(5)In my opinion, Table 5 contains Tables 6~9. Are Tables 6~9 necessary?

RESPONSE:

Table 5 reports the results of the model as applied in the multiclass classification task, whereas Tables 6-7 and 8-9 report the results of the model as applied in the single-class classification tasks for Or-01 and Or-05 respectively. Since these are separate results, we recommend that each of these tables remain included.

(6)In Section 4, it is better to specify the findings point-by-point.

RESPONSE:

Thanks for bringing this to our attention so we can improve the comprehension of the discussion. The first paragraph in section 4 was adjusted to list the key findings out point by point, and the subsequent section paragraphs were rearranged, edited, and one paragraph was added, to follow the listed findings in order.

(7)The authors just simply used the existing models, so the academic value is limited. It is better if the authors could improve the models.

RESPONSE:

The task of acquiring Real World Evidence from radiological reports is crucial for informing clinical decision making, and currently, its large-scale feasibility is impacted by the lack of appropriate computer systems to replace manpower. The study examines the feasibility for large-scale automatic interpretation of radiological reports using NLP, which is currently a limited application as most often human annotation is required for accurate report interpretation. Specifically, the study develops a BERT model based on the recently introduced OR-RADS standardized lexicon for radiological reporting. This is a novel NLP application, and the integration of the OR-RADS categories itself is an improvement on existing applications of the BERT model for report classification. For these reasons we believe the study presents important academic value.

Reviewer 2 Report

It is unclear how the True Label is retrieved? Is it from Cancer Registry, or who gave the True Label?

It is unclear how the 3 human radiologist was organized to predict labels for the test set. Was there double blind labeling to compare inter-rater agreement?

It is unclear what types of cancer are included in the dataset, and their distribution across train/validate/test subsets.

Line 223: Table 9 is for Test set

Author Response

Reviewer 2

INTRODUCTORY RESPONSE:

Thank you for your valuable comments on the manuscript. Our responses are addressed below.

It is unclear how the True Label is retrieved? Is it from Cancer Registry, or who gave the True Label?

RESPONSE:

The true labels are 1 of the 7 OR-RADS categories that were assigned to each radiological impression by board-certified attending radiologists at a tertiary cancer centre within an 11-month time period prior to the study experiments. The labels were assigned at the time of reporting. To improve the clarity of this, a statement at the end of paragraph 1 in section 2 was added to indicate what the “True Label” signifies.

It is unclear how the 3 human radiologist was organized to predict labels for the test set. Was there double blind labeling to compare inter-rater agreement?

RESPONSE:

Thank you for raising this point. The 3 human radiologists were assigned the prediction task with no overlap between cases. No inter-rater agreement was measured, but their accuracy was compared to the original OR-RADS assignment as described. This information was added to the end of section 2.

It is unclear what types of cancer are included in the dataset, and their distribution across train/validate/test subsets.

RESPONSE:

The data we had access to for this project did not include further information on cancer type, therapies, or other patient demographics as this information requires manual review to ensure accuracy. In the future, it may be best to optimize each BERT model to a single cancer subtype, for example lung or colorectal cancer, before applying it for Real World Evidence research. We have addressed this limitation in the discussion.

Line 223: Table 9 is for Test set

RESPONSE:

The table caption has been corrected.

Reviewer 3 Report

The paper is dedicated to such important task as processing of text reports or radiology scans by radiologists using natural language processing. While the task of annotating the scans is important, more important one is to interpret the results. Your research is important to establish guidelines for computer aided systems that allow doctors to process data with aid of computer system.

Remarks:

1. The main issue to address is the table size and its margins – table 1 don’t fit the margins. Hence, you need to resize the text somehow, using other terms that are abbreviations of the terms or use footnotes to describe them below on the page. I think the headers of the Table 1 should be shrunk to fit the page.

2. As I saw numerous times, you operate using the original, unaltered dataset, that shows impressions of the doctors. However, what I have noticed that it is imbalanced between the classes. While it won’t be possible to gather new information to complement the gaps, it would be interesting to see a synthetic dataset generated (or correctly said augmented) to address this problem. I think that it would be a good idea to test such a dataset in your future work.

3. Related to suggestion 2. Since the performance may be affected in imbalanced learning, it also affects the human performance as well. In the corresponding tables one can see that the human performance decreases drastically on less presented classes. In order to address this problem, I suggest to study human performance vs BERT performance on synthetic (augmented) dataset in your further studies.

Overall,  are quite supportive of the results since your study addresses the real world problem, uses the real world data (impressions or reports of medical imaging), that can be used to address some real problems in the radiology. Despite the minor issue with table 1, I don’t see serious problems in the paper, that would be worth noticing here. However, I want to address, that while your approach to data was purely methodologically correct (using original datasets on real world data), I see possible caveats on larger datasets, if the classes would be still unbalanced. 

I think that in the further research you can for sure rely upon BERT, but try to generate a synthetic dataset, which would allow 2 class and 7 class classification on the balanced dataset. I think in order to do so you need to establish either manual procedure for dataset augmentation or use some augmentation approach, using imbalanced learning methods like SMOTE or ADASYN, or use language transformer or other neural network model to generate samples, if it is possible.

Still, this is a suggestion for your further research and this is up to you to decide how to do it.

Author Response

Reviewer 3

INTRODUCTORY RESPONSE:

Thank you for your interest in the study and for sharing detailed comments towards advancing the research. Please find the following responses to your remarks. 

  1. The main issue to address is the table size and its margins – table 1 don’t fit the margins. Hence, you need to resize the text somehow, using other terms that are abbreviations of the terms or use footnotes to describe them below on the page. I think the headers of the Table 1 should be shrunk to fit the page.

RESPONSE:

Table 1 has been adjusted to fit within the text margins.

  1. As I saw numerous times, you operate using the original, unaltered dataset, that shows impressions of the doctors. However, what I have noticed that it is imbalanced between the classes. While it won’t be possible to gather new information to complement the gaps, it would be interesting to see a synthetic dataset generated (or correctly said augmented) to address this problem. I think that it would be a good idea to test such a dataset in your future work.

RESPONSE:

This is an interesting point to consider, thanks for raising it. We have added it to the list of future work expansions detailed in the last paragraph of section 4.

  1. Related to suggestion 2. Since the performance may be affected in imbalanced learning, it also affects the human performance as well. In the corresponding tables one can see that the human performance decreases drastically on less presented classes. In order to address this problem, I suggest to study human performance vs BERT performance on synthetic (augmented) dataset in your further studies.

RESPONSE:

Thank you for sharing this interesting observation about the human performance. In addition to referencing this expansion in the last paragraph of section 4, a comment on this observation was added to the 7th paragraph of section 4.

Overall,  are quite supportive of the results since your study addresses the real world problem, uses the real world data (impressions or reports of medical imaging), that can be used to address some real problems in the radiology. Despite the minor issue with table 1, I don’t see serious problems in the paper, that would be worth noticing here. However, I want to address, that while your approach to data was purely methodologically correct (using original datasets on real world data), I see possible caveats on larger datasets, if the classes would be still unbalanced. 

I think that in the further research you can for sure rely upon BERT, but try to generate a synthetic dataset, which would allow 2 class and 7 class classification on the balanced dataset. I think in order to do so you need to establish either manual procedure for dataset augmentation or use some augmentation approach, using imbalanced learning methods like SMOTE or ADASYN, or use language transformer or other neural network model to generate samples, if it is possible.

Still, this is a suggestion for your further research and this is up to you to decide how to do it.

RESPONSE:

Thank you for your support of the manuscript and study results and for the valuable suggestions for future work expansion.

Reviewer 4 Report

Referee report for Cancers - ”Applying natural language processing for single-report prediction of metastatic disease response using the OR-RADS lexicon”

The paper targets the adoption of natural language processing to classify clinical impressions from radiological reports. Reported performances are promising and motivate a standardization in reporting among radiologists in order to bound time consumption and costs. Unfortunately many useful details are missing, concerning for example the cancer types included in your study, as well as a more statistically stable analysis, like a cross validation.

The paper is not suitable for publication in Cancers in the current format. I recommended the following integrations in order to achieve a sufficient content.

1. in lines 20-21 in the Abstract there a repetition of a statement already used in the Simple Summary;

2. in line 25 you have to introduce the BERT acronym;

3. in the Introduction conclusion you should briefly summarize the performance improvement obtained by means of OR-RADS;

4. in lines 153-154 in Methods you describe the dataset partition in training, validation and test sets, instead multiple cross-validations are required to ensure performance stability;

5. in Section 3.1 you have to specify more precisely the comment for Table 1;

6. in Section 3.1.2, 3.1.3. and 3.2.2. a more complete analysis about the misclassifications is required by including information about e.g. cancer type, therapies ... in order to typify classification errors and compare such typifying features between BERT and humans;

7. in Table 9 are you referring to ”Test set”?

8. in lines 307-308 in Discussion the label Or01 should be referred to ”decreased”?

9. in lines 312-313 in Discussion the label Or05 should be referred to ”increased”?

10. in lines 324-325 the statement about non-disease-related comments should be verified by testing classification performances for a parametrized misbalance among training, validation and test sets;

11. the number of references is low.

Author Response

Reviewer 4

INTRODUCTORY RESPONSE:

Thank you for your thorough review of the manuscript and for raising important comments towards improving the content which are addressed below:

  1. in lines 20-21 in the Abstract there a repetition of a statement already used in the Simple Summary;

RESPONSE:

We would like to clarify that as per journal guidelines, they request a simple summary in lay terms in addition to the scientific abstract, hence the similarity.

  1. in line 25 you have to introduce the BERT acronym;

RESPONSE:

The BERT acronym is now defined in the abstract.

  1. in the Introduction conclusion you should briefly summarize the performance improvement obtained by means of OR-RADS;

RESPONSE:

Thank you for this important suggestion. The second paragraph in section 4 was expanded to explain the performance improvement in the BERT model obtained by the use of the OR-RADS compared to previous work applying BERT to a similar application using RECIST.

  1. in lines 153-154 in Methods you describe the dataset partition in training, validation and test sets, instead multiple cross-validations are required to ensure performance stability;

RESPONSE:

Thank you for this comment. Standard guidelines for reporting AI research, such as “Checklist for Artificial Intelligence in Medical Imaging (CLAIM)”, do not specify the need to study multiple splits, especially as this study serves primarily as proof of concept at this stage. Cross-validations will be an important consideration for future work. 

  1. in Section 3.1 you have to specify more precisely the comment for Table 1;

RESPONSE:

The Table 1 comment was expanded to clearly describe the contents of the table.

  1. in Section 3.1.2, 3.1.3. and 3.2.2. a more complete analysis about the misclassifications is required by including information about e.g. cancer type, therapies ... in order to typify classification errors and compare such typifying features between BERT and humans;

RESPONSE:

The data we had access to for this project did not include further information on cancer type, therapies, or other patient demographics. It is possible that the BERT model may fail more often to correctly predict for a subset of cancers, or a subset of therapies. While an investigation of potential effects of these variables on potential misclassifications by the BERT model would be a future potential area of research, we refrained from doing so given the relatively small sample size in this study and the low failure rate compared to the total number of variables that would be introduced by including cancer subtypes and therapies. In the future, it may be best to optimize each BERT model to a single cancer subtype, for example lung or colorectal cancer, before applying it for Real World Evidence research. We have addressed this limitation in the discussion.

  1. in Table 9 are you referring to “Test set”?

RESPONSE:

Thank you for picking up on this; the text has been updated.

  1. in lines 307-308 in Discussion the label Or01 should be referred to “decreased”?

RESPONSE:

The text has been corrected.

  1. in lines 312-313 in Discussion the label Or05 should be referred to “increased”?

RESPONSE:

The text has been corrected.

  1. in lines 324-325 the statement about non-disease-related comments should be verified by testing classification performances for a parametrized misbalance among training, validation and test sets;

RESPONSE:

Thank you for this comment. If this statement, “These non-disease-related comments likely misled the model’s prediction whilst the radiologists with domain expertise could interpret (and discard) them appropriately,” is what is being referred to, we would like to clarify that this is an observation of the impact of non-disease-related terms on model performance versus human performance. We have not conducted specific experiments to verify this, as this statement is speculative at this stage. We appreciate this suggestion, and we believe it can be explored in future work, as currently this study serves as proof-of-concept.

  1. the number of references is low.

RESPONSE:

We have added additional references to further support the text.

Round 2

Reviewer 1 Report

Accept.

None.

Author Response

Reviewer 1 (round 2):

Accept.

RESPONSE:

Thank you for accepting our revisions.

Reviewer 4 Report

Authors exhaustively replied to questions raised in my report.

Author Response

Reviewer 4 (round 2):

Authors exhaustively replied to questions raised in my report.

RESPONSE:

Thank you for accepting our revisions.